# Cultural Divergence in Psychedelic Use among Medical Students: An ESPAD-Adapted Survey among Poles and Iraqis

Ahmed Al-Imam [1,2,3,*] , Marek A. Motyka [4] and Michal Michalak [2]

1   Doctoral School, Poznan University of Medical Sciences, 61-806 Poznan, Poland
2   Department of Computer Science and Statistics, Poznan University of Medical Sciences, 61-806 Poznan, Poland; michal@ump.edu.pl
3   Department of Anatomy and Cellular Biology, College of Medicine, University of Baghdad, Baghdad 10047, Iraq
4   Institute of Sociological Sciences, University of Rzeszow, 35-959 Rzeszów, Poland; mmotyka@ur.edu.pl
*   Correspondence: ahmed.mohammed@comed.uobaghdad.edu.iq or tesla1452@gmail.com; Tel.: +48-(0)-733-598-132 or +964-(0)-771-433-8199

**Abstract:** Psychedelics can profoundly alter cognition and consciousness. Their use in Middle Eastern countries, including Iraq, is ambiguous. We aim to investigate psychedelic awareness and use among Iraqi and Polish medical students. We surveyed 739 university students from Poland (315) and Iraq (424) using 31 adapted questions from the European School Survey Project on Alcohol and Other Drugs (ESPAD). We conducted multivariable analyses based on binary logistic regression to identify the factors associated with psychedelic awareness and use. Most of the respondents were females (65.6%) and senior medical students (69.6%). Notably, the Polish students displayed a higher prevalence of psychedelic use (21.6% compared to 1.2%, $p < 0.001$), while the Iraqi participants exhibited a lower familiarity with psychedelics ($p < 0.001$). The multivariable model demonstrated a commendable level of statistical accuracy and satisfactorily conformed to the Hosmer–Lemeshow goodness-of-fit test (statistical accuracy = 91.61%, Nagelkerke $R^2$ = 0.488, $p$-value = 0.848). Several factors emerged as correlates of increased psychedelic use, including lacking an intact religious belief system or commitment to its practice (OR = 7.26, 95% CI [2.33, 22.60]). Additionally, those who perceived a low risk associated with sporadic psychedelic use (3.03, [1.44, 6.36]) were likelier to engage in such behavior. Other factors included residing in Poland (2.82, [0.83, 9.55]), maintaining positive attitudes toward psychedelics (2.49, [1.20, 5.14]), frequent social nightlife activities (2.41, [1.26, 4.61]), male gender (2.05, [1.10, 3.85]), and cigarette or tobacco smoking (2.03, [1.06, 3.86]). Significant disparities exist between Poles and Iraqis, influenced by religiosity, perceptions of the usage risks, geographical location, gender, attitudes toward psychedelics, parental leniency, and social freedom, especially concerning nightlife activities. Addressing the factors influencing psychedelic usage is paramount to responsible psychedelic engagement and culturally sensitive interventions to prevent misuse.

**Keywords:** addiction psychiatry research; ESPAD 2019; European School Survey Project on Alcohol and Other Drugs; public perception and awareness; risky behaviors; substance use disorders

## 1. Introduction

Psychedelics, also known as mystico-mimetics and entheogens, encompass a class of psychoactive substances that primarily act on the 5-HT$_{2A}$ receptors in the neurons of the central nervous system (CNS), particularly in the brain. Through this interaction, they can induce significant alterations in consciousness. These changes manifest as shifts in perception across sensory modalities, cognitive processes, emotions, and mood, often accompanied by vivid sensory and mystical experiences [1,2]. Notable examples of well-known psychedelic substances include LSD (lysergic acid diethylamide), psilocybin (derived from magic mushrooms), DMT (dimethyltryptamine), and mescaline (naturally

occurring in peyote and San Pedro cactus). Moreover, there are other substances, like ketamine, cannabis, and crystal methamphetamine, which are considered psychedelic analogs or quasi-psychedelics due to their potential to elicit psychedelic effects at specific doses [1,2]. The current study also seeks to serve as an informative resource for readers, shedding light on "mystical" psychedelics like DMT and the potent 5-MeO-DMT, colloquially known as the "God Molecule" [2].

Indeed, the available data on public awareness of these chemicals and their prevalence of use are limited, with substantial variations likely to exist based on factors like culture, ethnicity, religious affiliations, and geographical location. Notably, such disparities can be observed between developing nations, like those in the Middle East [3], and developed regions, such as Europe. These differences underscore the need for comprehensive research to understand the unique dynamics shaping the attitudes towards and behaviors related to these substances in diverse populations. A deeper exploration of these factors could provide valuable insights into formulating targeted public health interventions and educational campaigns to promote responsible recreational or medical use, when applicable (as with medical cannabis), and mitigate the potential risks associated with psychedelics.

According to the literature, the level of interest in psychedelics is not static and may fluctuate throughout the year [4]. A study conducted in 2022 revealed intriguing seasonal patterns of interest in psychedelics in Poland, with some fluctuations correlating with school breaks and holidays, while the emergence of the SARS-CoV-2 pandemic introduced additional complexities, causing shifts in temporal and spatial trends due to lockdown measures and changes in individual and societal behaviors, while the latter may relate to social anomie theory [5,6]. Notably, the authors identified that psilocybin and ayahuasca exhibited annual seasonality coinciding with the mushroom harvest season in Poland (September and October) [5]. Across Europe, there is a noticeable similarity in the seasonal trends in cannabis and psychedelics. Some of these patterns align with the mushroom harvesting season, while others coincide with holidays like Christmas, New Year's, and school vacations. The pandemic has also brought about considerable shifts in the online interest surrounding these substances in some nations in the European Union and the United Kingdom [7].

On another note, researchers have dedicated considerable efforts to studying risky behaviors encompassing substance abuse and addiction to various substances such as tobacco, alcohol, cannabis, CNS depressants (including opioids), psychostimulants, neuroleptics, over-the-counter drugs (OTC), illicit (regulated) research chemicals, and other substances. Examining risky behaviors is paramount to psychiatrists and social workers, as they can lead to potentially dangerous or reckless outcomes, such as accidents, homicide, and the (para)suicide spectrum (suicide ideation, parasuicide, and suicide) [8,9]. However, when it comes to well-designed and reliable studies on substance misuse behaviors among high school and college students, including medical students, such research is scarce, particularly in the developing world, including Iraq.

Over the past two decades, Iraq has experienced a concerning and exponential increase in the use of illicit drugs, encompassing psychotropic medications, antipsychotics, anxiolytics, antidepressants, and, most notably, psychostimulants. However, data are scarce concerning the prevalence of hallucinogenic drug use among specific populations of interest, including medical students in Iraq [3] and other Middle Eastern countries, particularly following the "Arab Spring" movement.

On the other hand, hallucinogenic drug and psychedelic use prevalence exhibits substantial variability across different countries and regions. Nevertheless, according to the United Nations Office on Drugs and Crime (UNODC) World Drug Report 2021, the global prevalence of hallucinogenic drug use, including substances like LSD, psilocybin, and peyote, among adults aged 15–64 in 2019, was estimated to be approximately 0.3% [10]. As the issue of drug use continues to grow, policymakers, public health authorities, and researchers must gather more comprehensive data to understand the extent of hallucino-

genic drug use in Iraq and the broader Middle Eastern context. Such data can inform the regulatory authorities to promote public health and well-being.

It is crucial to underscore that in Iraq, the abuse of certain substances, particularly psychostimulants, cannabis, opioids, and over-the-counter drugs (OTC), has increased, especially after 2003, due to the social-cultural changes and political unrest following the American invasion that resulted in the removal of Saddam Hussein from power. Potent psychostimulants, such as amphetamine and amphetamine-type stimulants (ATS), are the most prevalent, with dextroamphetamine and methamphetamine being the primary amphetamines available in the country, as reported by Chachan and Al-Hemiary (2023) [11]. Among these, crystal methamphetamine, commonly known as "crystal" or "crystal meth", holds a significant presence in Iraq's illicit drug scene [11,12]. Al-Hemiary et al. (2014) corroborated this trend, demonstrating an exponential increase in the illicit use of ATS, methamphetamine, and captagon, particularly in the governorate of Al-Basrah [13].

Although there are limited available data on the prevalence of hallucinogenic drug use in Iraq and the Middle East, the United Nations Office on Drugs and Crime (UNODC) 2021 World Drug Report indicated that the use of hallucinogenic drugs and psychedelics in the Middle East and North Africa is relatively low compared to other regions, such as Europe, with an estimated prevalence of less than 0.1% among adults aged 15–64 [10]. The low prevalence of psychedelic use could be related to ethnocultural, religious, or other factors, which are principally attributed to drug trafficking, black market demands, and the availability of certain illicit substances in the Middle East and neighboring nations.

Various studies have attempted to estimate the prevalence of psychedelic use. However, the reliability of some of these studies is questionable due to the absence of externally validated survey tools to assess the phenomenon reliably, particularly among adolescents and college students. In contrast, the European School Survey Project on Alcohol and Other Drugs (ESPAD) stands out as a reliable and reputable tool. As officially stated by the ESPAD project, "The ESPAD Report 2019 features information on students' experience of, and perceptions about, a variety of substances, including tobacco, alcohol, illicit drugs, inhalants, pharmaceuticals, and new psychoactive substances (NPS). Social media use, gaming, and gambling are also covered." [14].

### 1.1. Study Importance and Novelty

The current study addresses a pressing societal concern at a global scale, encompassing Iraq and Poland in particular. By utilizing a survey adapted from the reliable ESPAD survey, we aim to examine the awareness and usage of psychedelics thoroughly. This study also explores the potential use of psychedelics to enhance academic performance. In the forthcoming discussion, we aim to elaborate on the relevant conceptual framework, delving into how microdosing and psychedelic use intersect as tools for cognitive enhancement. The research could also help understand the barriers for young women with drug addictions and psychedelics. Marek et al.'s (2022) study found that Polish females faced limited support and stigma-related challenges [15,16].

### 1.2. Study Aims and Rationale

Our main goal is to assess the awareness and usage of psychedelic substances among medical students from Poland and Iraq. This study aims to uncover differences in the prevalence and understanding of psychedelic substance use, shedding light on the cultural, ethnic, religious, and educational factors influencing these behaviors. We hypothesize that Polish medical students have higher awareness and usage rates than their Iraqi counterparts.

As a secondary objective, this study provides culturally based insights that could guide managing intoxication or adverse events related to widespread psychedelic overdosing. Psychedelic overdosing presents a significant public health concern in cultures worldwide, including Poland and Iraq, due to its associated health risks, such as psychosis and cardiovascular issues. Contributing factors include limited education on psychedelics' risks, varying legal statuses leading to misinformation and clandestine distribution, cul-

tural attitudes affecting substance use perceptions and treatment-seeking behaviors, and disparities in access to healthcare and treatment services. Addressing this issue requires comprehensive approaches encompassing public education, harm reduction strategies, enhanced access to treatment, and culturally sensitive policies to effectively mitigate the impact of psychedelics overdosing across diverse cultural contexts.

## 2. Materials and Methods

### 2.1. Research Ethics

The study did not require approval from the bioethics committee. On another note, the current study strictly adhered to the guidelines outlined in the Declaration of Helsinki by the World Medical Association and follows the ethical principles of the Framingham Consensus of 1997. Before participation, informed consent was obtained from each participant. While no approval from the bioethics committee was necessary, it underwent examination by the Institutional Review Board (Ethics Committee) of Poznan University of Medical Sciences (Uniwersytet Medyczny im. Karola Marcinkowskiego w Poznaniu). It was exempted from approval requirements on 12 November 2021 due to its observational (non-experimental) nature.

### 2.2. Sampling and Data Collection

The present study adopted a cross-sectional design and convenience sampling for participant selection. The survey distribution was facilitated by the university mailing lists of the University of Baghdad (and other universities in Baghdad, Iraq) and Poznan University of Medical Sciences (the only medical university in Poznan, Poland). The survey was distributed to 3768 students, comprising 1306 Poles and 2462 Iraqis. However, only 739 students responded and completed the survey, with 315 respondents from Poland and 424 from Iraq. The primary focus for the study was on medical students in Iraq and Poland. Consequently, surveys were available in Polish, Arabic, and English (Supplementary Materials: Table S1). Emphasizing confidentiality, the survey was anonymous and did not collect any personal identifiers from the respondents.

### 2.3. The European School Survey Project on Alcohol and Other Drugs 2019

The survey design was grounded in the ESPAD 2019 framework, constituting the seventh iteration of data collection efforts by the ESPAD authorities. Notably, the ESPAD 2019 initiative engaged nearly 100,000 students from 35 countries in 2019 [14], providing a robust foundation for our study. By drawing upon this established framework, our survey sought to capture relevant and comparable data, enhancing the reliability and validity of our findings.

We gauged participants' awareness by analyzing their responses to survey items 19, 20, and 28 regarding the definition of psychedelics, knowledge on such substances, and awareness of their adverse events, respectively (Supplementary Materials: Table S1). Accurate responses on psychedelic substances were rewarded with a positive point (+1), while failing to select the correct substances incurred a penalty of minus one point (−1). Conversely, opting for an erroneous substance (non-psychedelic) resulted in a deduction of one point (−1). On the flip side, refraining from choosing non-psychedelic substances garnered a point (+1). The ultimate (summative) score, denoting one's knowledge about psychedelics, encompassing familiarity with well-known varieties, is designated as "Score A". A similar method was used to assess comprehension of the adverse effects of psychedelics, and we denoted the corresponding score as "Score B". Consequently, the aggregate (total) score, derived from the summation of the preceding two individual scores, serves as an indicator of one's comprehension of the definition of these chemical compounds, their acquaintance with commonly encountered psychedelics, and their awareness of adverse effects.

The former scoring criteria facilitated the creation of an ordinal scale score, for which the median was computed. Subsequently, participants were categorized based on whether they exceeded (>median) or fell at or below the median score (≤median). The evaluation

procedure (scoring) was carried out by two assessors (raters) who worked independently and were unaware of each other's evaluations (employing a double-masked methodology). In cases where inconsistencies in the scoring computation arose, a third neutral individual (rater) was engaged to address any divergences between the initial two raters.

Further, students were tasked with expressing their viewpoints on psychedelics by answering question 27 of the survey. They were allowed to provide open-ended comments, which underwent subsequent natural language processing (NLP) using Python's PyTorch-NLP library. The NLP outcome encompassed categorizations of "positive", "negative", or "neutral", effectively encapsulating the broader stance of each student on psychedelics and their utilization.

### 2.4. Respondents' Attributes and ESPAD's Psychedelic and Quasi-Psychedelic Domains

The survey comprised two distinct sections. The first part (questions 1 to 17) focused on gathering data related to socio-demographics, religious practices (or lack thereof), financial status, academic performance, school attendance, daily lifestyle, involvement in social and cultural activities, parents' educational background, relationship with parents, family dynamics, and tobacco and alcohol consumption among the respondents. In the second part (questions 18 to 31), data were collected explicitly concerning cannabis and psychedelics, with the former being a pertinent quasi-psychedelic substance. The objective was to determine respondents' awareness of psychedelic substances and the extent of their usage. The key points addressed in this section included understanding the correct definition of psychedelic substances, recognizing different psychedelic drugs, assessing the frequency and duration of psychedelic use, evaluating the challenges associated with accessing or acquiring these substances, and exploring opinions on adverse effects.

### 2.5. Statistical Analysis and Methods

For data analysis, the research team compiled the gathered data using Microsoft Excel 2016 and transferred them into TIBCO Statistica version 13.3 (2018) for further data preparation and analysis. Based on binary logistic regression, multivariable analyses were employed to accomplish two objectives: (1) identifying potential predictors of higher awareness scores among Iraqis and Poles and (2) pinpointing the possible risk factors associated with psychedelic use. The findings were presented as odds ratios (ORs) and their corresponding 95% confidence intervals (95% CIs). The goodness of fit of the logistic regression models was assessed through the Hosmer–Lemeshow test (HL test). Nagelkerke's $R^2$ was reported as an indicator of the proportion of the total variation in the dependent variable that the independent variables could explain.

## 3. Results

### 3.1. The Prevalence of Psychedelic Use

The study encompassed 739 participants who completed the survey, including 424 from Iraq and 315 from Poland. Among the Iraqi participants, 1.18% (5 out of 424) reported using psychedelics. In contrast, among the Polish participants, the prevalence of psychedelic use was notably higher, at 21.59% (68 out of 315). A significant difference in psychedelic use was evident, favoring Poles, as indicated by the $\chi^2$ test (OR = 23.07, 95% CI [9.18, 57.99], $p < 0.001$).

### 3.2. Socio-Demographic and Socio-Cultural Differences

There were significant differences between the Polish and Iraqi medical students concerning many parameters (Supplementary Materials: Table S2). Concerning the first part of the survey, significant disparities emerged concerning the responses of the Poles and Iraqis across several questions. Notably, the questions with the highest odds ratio according to the Chi-square test (ranked in descending order) encompassed alcohol consumption, religious convictions (religious beliefs and practices), internet usage for leisure, engagement in sports, parental awareness of their son's (or daughter's) whereabouts during weekend

nights, age, parental inclination to establish rules for their son (or daughter) beyond the household, evening social outings (social nightlife activities), engagement in cultural book reading, and deliberate absenteeism from university classes.

Poles exhibited a markedly greater proclivity toward alcohol consumption (OR = 180.44, 95% CI [100.17, 325.04], *p* < 0.001), while Iraqis tended to have more prevalent religious beliefs and commitment to relevant practices (OR = 18.01, [12.44, 26.08], *p* < 0.001). Conversely, Poles demonstrated a higher tendency to engage in using the internet for leisure (OR = 13.92, [1.85, 104.843], *p* = 0.001) and participate more in sports (OR = 10.97, [6.07, 19.82], *p* < 0.001), whereas Iraqi parents showed greater awareness of their son's (or daughter's) whereabouts during weekend nights (OR = 6.63, [3.47, 12.66], *p* < 0.001) (Figure 1).

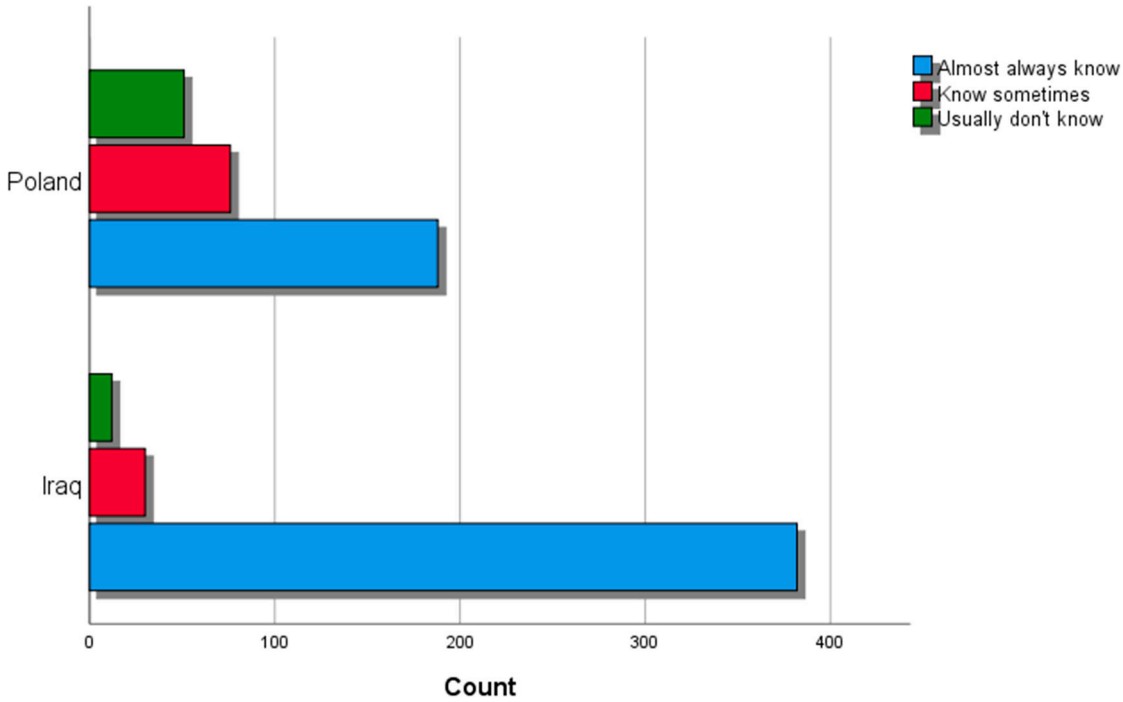

**Figure 1.** Whereabouts awareness among Poles and Iraqis.

On the other hand, Iraqi parents appeared more inclined to establish clear rules for their sons (or daughters) outside the home (OR = 6.17, [4.46, 8.48], *p* < 0.001). In contrast, the Polish students participated in more social nightlife activities, which were often cryptic to their parents (OR = 5.35, [3.27, 8.77], *p* < 0.001). Notably, Poles were more enthusiastic about reading cultural books (OR = 5.76, [3.49, 9.49], *p* < 0.001), while Iraqis tended to miss university classes deliberately (OR = 5.68, [4.07, 8.00], *p* < 0.001).

Following the 2003 invasion of Iraq, various factors could have contributed to Iraqis deliberately missing school attendance. These include security concerns stemming from the conflict, displacement, and infrastructure damage disrupting schooling, economic hardship hindering families' ability to afford education-related expenses, psychological trauma from violence, and socio-cultural factors such as gender discrimination.

Regarding the second part of the survey, notable disparities surfaced concerning the responses of Polish and Iraqi medical students across various questions. Particularly striking were the questions with the highest odds ratio, as determined using the Chi-square test, listed here in descending order: proficiency in accurately defining psychedelic drugs, engagement in psychedelic use, using psychedelics as academic performance enhancers, prior knowledge of psychedelics, and views on the ease of access to psychedelics.

In terms of the former parameters, Polish medical students demonstrated notable advantages. For instance, a significant proportion of Polish students were able to define

psychedelics accurately (OR = 42.37, [19.56, 91.78], *p* < 0.001), whereas a significantly smaller number of their Iraqi counterparts reported using these substances (23.07, [9.18, 57.99], *p* < 0.001). Furthermore, a higher percentage of Polish students utilized psychedelics as cognitive enhancers for better academic performance (OR = 12.44, [1.57, 98.71], *p* < 0.003). Additionally, many Polish students perceived the ease of accessing or purchasing psychedelic substances (OR = 2.16, [1.59, 2.94], *p* < 0.001). In contrast, more Iraqi students held pessimistic (opposing) views regarding psychedelics (OR = 1.51, [1.13, 2.02], *p* = 0.006), deeming even their sporadic use as risky (OR = 1.33, [0.99, 1.79], *p* = 0.054) (Figure 2).

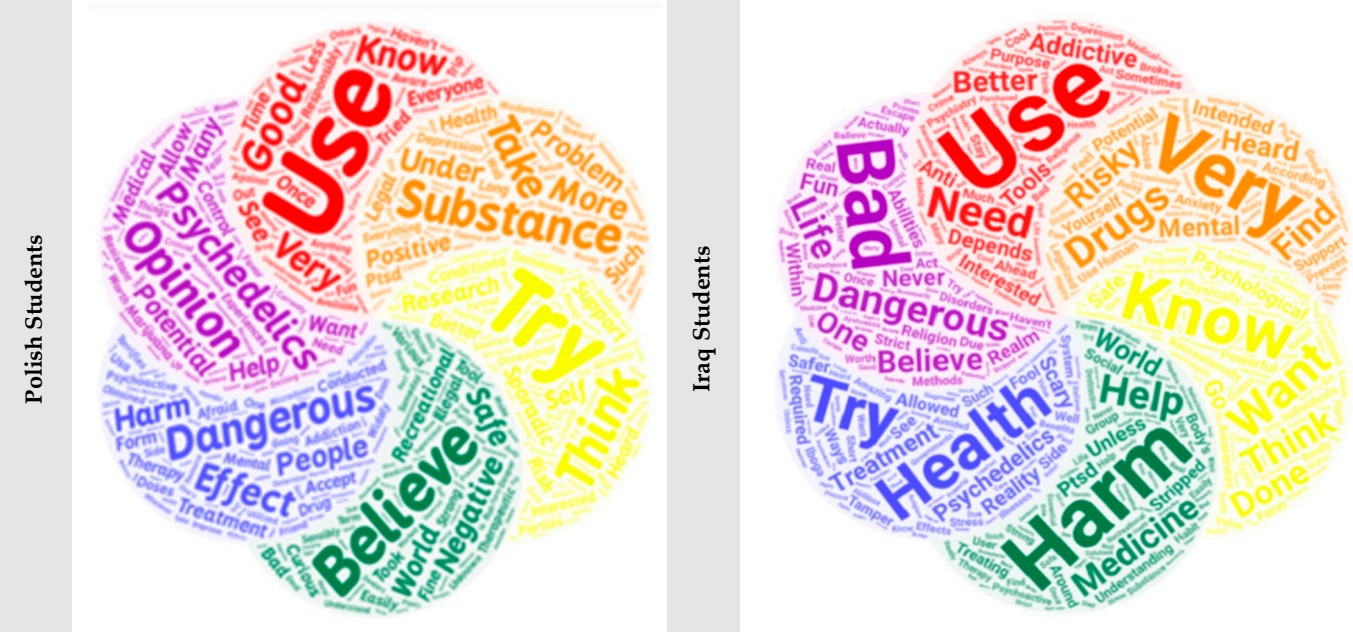

**Figure 2.** Word cloud showing medical students' opinions on psychedelics.

The preceding analysis has unveiled profound distinctions between Polish and Iraqi medical students. Polish students are more prone to alcohol consumption, using the internet for leisure activities, actively participating in sports, and partaking in vibrant social nightlife activities. Additionally, they exhibit a heightened interest in cultural literature readings and explore the utilization of psychedelics for cognitive enhancement. Conversely, their Iraqi counterparts manifest more pervasive religious beliefs, adhere to stricter parental regulations, and possess limited familiarity with psychedelics. These disparities are crucial in elucidating the variations in the awareness and use of psychedelic substances.

*3.3. Psychedelic Awareness Scores among Poles and Iraqis*

Regarding the entire sample, we calculated the range, the median, and the interquartile range (IQR) for the total score (range: −11 to 24) (median = 7.92, IQR = [2.10, 14.32]), score A (−7 to 10) (2.15, [−1.63, 6.42]), and score B (−8 to 14) (6.23, [2.43, 9.31]). A noteworthy observation is that, when compared to the Polish participants, those from Iraq exhibited lower scores across all categories of psychedelics-related knowledge, as reflected in the median values for score A (−0.89 vs. 6.64), score B (3.89 vs. 8.55), and the total score (3.20 vs. 14.75). The difference was significant across each scoring category, indicating higher awareness among Poles, as confirmed using the Mann–Whitney U test (*p*-value < 0.001). The previous observation could also explain the significantly lower prevalence of psychedelic use among Iraqi students compared to their Polish counterparts. On the other hand, there were no significant score differences between students of medicine and those studying dentistry and pharmacy, whether in Iraq or Poland.

### 3.4. Knowledge of Psychedelics across Other Survey Parameters

Knowledge about psychedelics displayed variations across the survey's questions and parameters (Table 1). These differences proved highly significant, with consistently low p-values ($p < 0.001$). Given the ordinal scale of the total knowledge score, our analysis employed non-parametric univariable models, such as the Kruskal–Wallis H and Mann–Whitney U tests. A significant disparity emerged when examining the relationship between religious beliefs and psychedelic knowledge. Medical students who did not adhere to any religious belief system or did not actively practice that system exhibited significantly greater knowledge about psychedelics compared to their counterparts who adhered to and practiced a particular belief system (median = 13, IQR [9, 17] vs. median = 4, IQR [1, 9], $p < 0.001$). Similarly, students engaged in athletic activities demonstrated a higher familiarity with psychedelics. Post hoc pair comparisons confirmed the presence of significant differences among all three subgroups, with the most pronounced effect size observed between those who never participated in sports and those who engaged in sports daily or weekly (4, [1, 8] vs. 11, [4, 16], $p < 0.001$). Interestingly, the educational level of students' mothers exhibited a significant disparity in the scores between those with primary school education (or less) and those with secondary school education or higher (3, [−1, 9] vs. 8, [3, 15], $p < 0.001$). On another note, significant variations in scores were evident between students who were dissatisfied about their relationship with their fathers and those who reported a satisfactory relationship (12, [7, 17] vs. 7, [1, 13], $p < 0.001$).

**Table 1.** Knowledge of psychedelics across different survey parameters.

| Variable | | Median | Interquartile Range | | p-Value |
|---|---|---|---|---|---|
| | | | Lower Bound | Upper Bound | |
| Age | ≤20 years (junior students) | 5.00 | 1.00 | 8.50 | **<0.001** |
| | >20 years (senior students) | 9.50 | 3.00 | 15.00 | |
| Religious status | Present | 4.00 | 1.00 | 9.00 | **<0.001** |
| | Absent | 13.00 | 9.00 | 17.00 | |
| Country | Poland | 15.00 | 11.00 | 18.00 | **<0.001** |
| | Iraq | 3.00 | −0.75 | 7.00 | |
| Residence | Urban | 7.00 | 2.00 | 14.00 | 0.057 |
| | Rural | 10.50 | 4.75 | 15.00 | |
| Academic performance (in the last semester) | Low | 8.00 | 2.50 | 13.50 | **<0.001** |
| | Average | 7.00 | 1.00 | 14.00 | |
| | High | 11.00 | 4.00 | 15.00 | |
| Missing classes (because of illness) | None | 10.00 | 3.00 | 15.00 | **<0.001** |
| | A few days | 5.00 | 1.00 | 10.00 | |
| | Several days | 3.00 | −1.00 | 9.25 | |
| Missing classes (intentional) | None | 11.00 | 4.00 | 15.00 | **<0.001** |
| | A few days | 5.00 | 1.00 | 12.00 | |
| | Several days | 5.00 | 1.00 | 9.00 | |
| Missing classes (other reasons) | None | 4.00 | 10.00 | 15.00 | **<0.001** |
| | A few days | 6.00 | 1.00 | 12.00 | |
| | Several days | 3.00 | 0.25 | 7.50 | |
| Participating in sports | Never | 4.00 | 1.00 | 8.00 | **<0.001** |
| | Several times a year | 7.00 | 1.25 | 14.00 | |
| | On a daily or weekly basis | 11.00 | 4.00 | 16.00 | |
| Reading cultural books | Never | 4.00 | −1.00 | 8.00 | **<0.001** |
| | Several times a year | 7.00 | 1.75 | 14.25 | |
| | On a daily or weekly basis | 12.00 | 5.75 | 15.00 | |

**Table 1.** *Cont.*

| Variable | | Median | Interquartile Range | | *p*-Value |
|---|---|---|---|---|---|
| | | | **Lower Bound** | **Upper Bound** | |
| Going out in the evening | Never | 4.50 | 0.25 | 9.00 | **<0.001** |
| | Several times a year | 9.00 | 3.00 | 15.00 | |
| | On daily or weekly basis | 9.00 | 2.25 | 14.00 | |
| Playing an instrument (or other hobbies) | Never | 5.00 | 1.00 | 9.00 | **<0.001** |
| | Several times a year | 9.00 | 2.00 | 15.00 | |
| | On a daily or weekly basis | 11.00 | 5.00 | 15.00 | |
| Going around (with friends) | Never | 11.00 | 0.00 | 6.00 | **0.019** |
| | Several times a year | 8.00 | 2.25 | 15.00 | |
| | On a daily or weekly basis | 8.50 | 2.25 | 14.00 | |
| Using the internet (for leisure activities) | Never | 1.00 | −3.00 | 6.00 | **<0.001** |
| | Several times a year | 2.50 | −1.00 | 7.75 | |
| | On a daily or weekly basis | 9.00 | 3.00 | 15.00 | |
| Mother's educational level | ≥Secondary school | 8.00 | 3.00 | 15.00 | **<0.001** |
| | ≤Primary school | 3.00 | −1.00 | 9.00 | |
| Parents set definite rules (inside home) | Almost always | 5.00 | 0.00 | 11.00 | **<0.001** |
| | Sometimes | 9.00 | 2.00 | 15.00 | |
| | Almost never | 9.00 | 3.00 | 15.00 | |
| Parents set definite rules (outside home) | Almost always | 4.00 | 0.00 | 8.00 | **<0.001** |
| | Sometimes | 6.00 | 1.00 | 12.00 | |
| | Almost never | 12.00 | 6.00 | 17.00 | |
| Parents know (whom I am with during the evenings) | Almost always | 5.00 | 1.00 | 11.00 | **<0.001** |
| | Sometimes | 11.50 | 5.00 | 17.00 | |
| | Almost never | 14.00 | 8.00 | 17.00 | |
| Parents know (where I am in the evenings) | Almost always | 5.00 | 1.00 | 11.00 | **<0.001** |
| | Sometimes | 11.50 | 5.00 | 17.00 | |
| | Almost never | 14.00 | 8.00 | 17.00 | |
| I can easily borrow money (from parents) | Almost always | 7.00 | 2.00 | 13.00 | **<0.001** |
| | Sometimes | 11.50 | 2.75 | 16.00 | |
| | Almost never | 10.00 | 5.00 | 17.00 | |
| I can easily get money as a gift (from parents) | Almost always | 7.00 | 2.00 | 13.00 | **0.008** |
| | Sometimes | 10.00 | 2.75 | 15.00 | |
| | Almost never | 7.00 | 2.75 | 12.00 | |
| My family really tries to help me | Agree | 8.00 | 3.00 | 15.00 | 0.057 |
| | Neutral | 6.00 | 1.00 | 11.00 | |
| | Disagree | 8.00 | 1.00 | 15.00 | |
| Parents know (where I spend weekend nights) | Almost always | 7.00 | 1.00 | 13.00 | **<0.001** |
| | Sometimes | 13.00 | 6.00 | 18.00 | |
| | Almost never | 14.00 | 8.00 | 17.00 | |
| Satisfaction (relationship with father) | Satisfied | 7.00 | 1.00 | 13.00 | **<0.001** |
| | Neither satisfied/dissatisfied | 10.00 | 4.00 | 15.00 | |
| | Not satisfied | 12.00 | 7.00 | 17.00 | |
| Smoking cigarettes or tobacco | Yes | 3.00 | 11.00 | 16.00 | **0.008** |
| | No | 2.00 | 7.00 | 14.00 | |
| Drinking alcohol | Often | 14.50 | 11.00 | 18.00 | **<0.001** |
| | Sometimes | 15.00 | 11.00 | 19.00 | |
| | Seldom | 13.00 | 8.50 | 16.00 | |
| | No | 3.00 | 0.00 | 8.00 | |

**Table 1.** *Cont.*

| Variable | | Median | Interquartile Range | | *p*-Value |
|---|---|---|---|---|---|
| | | | Lower Bound | Upper Bound | |
| Have you ever used psychedelics? | Yes, in the last 30 days | 14.50 | 11.75 | 17.00 | **<0.001** |
| | Yes, in the last 12 months | 17.00 | 13.00 | 19.00 | |
| | Yes, more than 12 months ago | 17.00 | 13.00 | 20.00 | |
| | Never | 7.00 | 1.00 | 13.00 | |
| Difficulty of getting psychedelics | Easy | 14.00 | 8.00 | 17.00 | **<0.001** |
| | Don't know | 6.00 | 1.00 | 12.00 | |
| | Difficult | 6.00 | 1.00 | 12.00 | |
| Difficulty of getting cannabis | Easy | 14.00 | 10.00 | 17.00 | **<0.001** |
| | Don't know | 5.00 | 1.00 | 10.00 | |
| | Difficult | 3.00 | 0.00 | 8.00 | |
| Psychedelics, risk of using once or twice | No risk | 6.00 | 1.00 | 13.00 | **<0.001** |
| | Low risk | 11.50 | 5.00 | 17.00 | |
| | High risk | 7.00 | 2.00 | 14.00 | |
| Psychedelics, risk of using occasionally | No risk | 5.00 | 0.00 | 12.75 | **<0.001** |
| | Low risk | 10.00 | 4.50 | 16.00 | |
| | High risk | 8.00 | 3.00 | 15.00 | |
| Psychedelics, risk of using regularly | No risk | 5.00 | −1.00 | 11.00 | **<0.001** |
| | Low risk | 7.00 | 1.00 | 13.25 | |
| | High risk | 9.00 | 3.00 | 15.00 | |
| Psychedelic use as a cognitive enhancer | Yes | 16.50 | 11.25 | 17.50 | **0.005** |
| | No | 8.00 | 2.00 | 14.00 | |
| Opinion on psychedelics | Positive | 15.00 | 9.50 | 18.00 | **<0.001** |
| | Neutral | 5.00 | 0.00 | 12.00 | |
| | Negative | 7.00 | 3.00 | 13.00 | |

The *p*-value relates to the total score of psychedelic awareness. Significant *p*-values are in bold font. The statistical analysis was based on non-parametric univariable models: Kruskal–Wallis H and Mann–Whitney U tests.

Medical students whose parents imposed stringent and consistently enforced rules regarding their activities outside their homes tended to exhibit lower levels of awareness. Notably, the post hoc testing revealed a difference between those whose parents almost always set rules and those whose parents rarely did so (4, [0, 8] vs. 12, [6, 17], *p* < 0.001). Similarly, individuals who engaged in smoking or alcohol consumption demonstrated a superior knowledge concerning psychedelics and their adverse effects. For example, those who refrained from alcohol had significantly lower awareness scores compared to those who consumed it regularly (3, [0, 8] vs. 14.5, [11, 18], *p* < 0.001). Further, divergent perspectives on psychedelics also yielded notably distinct awareness scores, particularly among those with positive versus negative opinions on psychedelics (15, [9.5, 18] vs. 7, [3, 13], *p* < 0.001). In contrast, medical students who perceived sporadic psychedelic use as low-risk consistently achieved higher scores than those who perceived it as high-risk (11.5 [5, 17] vs. 7, [2, 14], *p* < 0.001). As anticipated, individuals with psychedelic use experience exhibited notably higher scores than non-users. The most substantial difference was observed when comparing non-users to those who used psychedelics for at least one year (7, [1, 13] vs. 17, [13, 20], *p*-value < 0.001), indicating that "veterans" or chronic psychedelic users usually are genuinely knowledgeable concerning psychedelics.

### 3.5. Factors Influencing a Better Understanding of Psychedelics

A multivariable analysis, based on binary logistic regression, was used to identify medical students with a better understanding of psychedelics (≤median vs. >median) concerning their total score. The model possessed good statistical accuracy and fulfilled the requirement of the Hosmer–Lemeshow goodness-of-fit test (accuracy = 81.60%, Nagelkerke

$R^2 = 0.558$, *p*-value = 0.193). Furthermore, the model exhibited robustness concerning its statistical accuracy and predictive values (PPV = 80.45%; NPV = 82.44%; sensitivity = 76.99%; specificity = 85.23%). The factors promoting higher psychedelic awareness were identified (Table 2), including religious status, parents' strictness about rules at home, risk perception of regular use of psychedelics, family's openness to engaging in discussing their son's (or daughter's) problems, gender, and smoking.

**Table 2.** Multivariable analysis of factors associated with higher psychedelic awareness.

| Parameter | | *p*-Value | Odds Ratio | 95% Confidence Interval | |
|---|---|---|---|---|---|
| | | | | Lower Bound | Upper Bound |
| Religious status | Absent | <0.001 | **6.63** | 3.90 | 11.26 |
| | Present | | | | |
| Have you heard about psychedelics before? | Yes | <0.001 | **2.85** | 1.60 | 5.01 |
| | No | | | | |
| My parents set definite rules about what I can do at home | Almost never | <0.001 | **2.34** | 1.50 | 3.66 |
| | Sometimes | 0.023 | **1.97** | 1.10 | 3.53 |
| | Almost always | | | | |
| Risk perception of psychedelic use (regular use) | Low risk | 0.009 | **2.06** | 1.20 | 3.56 |
| | No risk | 0.090 | **1.81** | 1.05 | 3.31 |
| | High risk | | | | |
| I can talk about my problems with my family | Agree | 0.652 | 1.17 | 0.60 | 2.28 |
| | Disagree | 0.065 | **1.91** | 0.96 | 3.78 |
| | Neutral | | | | |
| Gender | Male | 0.008 | **1.81** | 1.17 | 2.80 |
| | Female | | | | |
| Smoking | No | 0.066 | **1.67** | 0.97 | 2.89 |
| | Yes | | | | |

For multiple categories, the last one is the reference. Odds ratios with significant *p*-values are in bold font.

The analysis yielded significant findings regarding medical students' knowledge of psychedelics. Notably, students without religious beliefs or those not practicing a religious belief system displayed a superior understanding of psychedelics (OR = 6.63, 95% CI [3.90, 11.26], *p* < 0.001). Similarly, prior exposure to information about psychedelics was associated with higher scores (2.85, [1.60, 5.01], *p* < 0.001). Conversely, strict parental rules at home were associated with reduced knowledge (2.34, [1.50, 3.66], *p* < 0.001). Those perceiving a low risk (or no risk) of regular psychedelic use were more knowledgeable (2.06, [1.20 to 3.56], *p* < 0.009 and 1.81, [1.05, 3.31], *p* < 0.090, respectively). Students who struggled to discuss problems with their families also exhibited superior scores (1.91, [0.96, 3.78], *p* = 0.065). Males (1.81, [1.17, 2.80], *p* = 0.008) and cigarette/tobacco users (1.67, [0.97, 2.89], *p* = 0.066) were more knowledgeable. These findings shed light on the complex factors influencing medical students' awareness of psychedelics across diverse cultural contexts.

In summary, the predominant factors influencing a superior awareness concerning psychedelics, as highlighted by their highest odds ratios, signify an increased likelihood of medical students better understanding psychedelics when they do not identify with any specific religious belief or its practice, regardless of the particular religious orientation. Additionally, parents' leniency in setting rules at home and a perceived lower risk associated with regular psychedelic use play crucial roles in shaping students' comprehension of these substances. Notably, Iraqi medical students derive their knowledge more from being aware of the adverse events linked to psychedelic use, while their Polish counterparts appear to be less influenced by information concerning the perceived risk associated with using psychedelics.

*3.6. Risk Factors for Psychedelic Use*

A comprehensive multivariable analysis, based on binary logistic regression, was conducted on the entire sample to discern the potential risk factors associated with psychedelic use. The model demonstrated a high level of statistical accuracy and successfully met the criteria of the Hosmer–Lemeshow goodness-of-fit test (statistical accuracy = 91.61%, Nagelkerke $R^2$ = 0.488, *p*-value = 0.848). Furthermore, the multivariable model exhibited reasonable predictive values, encompassing a positive predictive value (PPV) of 62.22% and a negative predictive value (NPV) of 93.52%, all stemming from the identification of seven significant risk factors (Table 3 and Figure 3), including religious status, perceived risk of sporadic use of psychedelics, geographic location (Poland or Iraq), opinion on psychedelics, frequency of going out in the evenings (nightlife), gender, and smoking.

**Table 3.** Multivariable analysis of risk factors of psychedelic use.

| Parameter | | *p*-Value | Odds Ratio | 95% Confidence Interval | |
|---|---|---|---|---|---|
| | | | | Lower Bound | Upper Bound |
| Religious status | Absent | 0.001 | **7.26** | 2.33 | 22.60 |
| | Present | | | | |
| Risk perception of psychedelic use (sporadic use) | Low risk | 0.003 | **3.03** | 1.44 | 6.36 |
| | No risk | 0.006 | **3.01** | 1.37 | 6.63 |
| | High risk | | | | |
| Geographic location | Poland | 0.095 | **2.82** | 0.83 | 9.55 |
| | Iraq | | | | |
| Opinion on psychedelics | Positive | 0.014 | **2.49** | 1.20 | 5.14 |
| | Neutral | 0.495 | 1.34 | 0.58 | 3.13 |
| | Negative | | | | |
| Frequency of going out in the evening | Daily | 0.008 | **2.41** | 1.26 | 4.61 |
| | Several times a year | 0.955 | 0.94 | 0.11 | 7.88 |
| | Never | | | | |
| Gender | Male | 0.025 | **2.05** | 1.10 | 3.85 |
| | Female | | | | |
| Smoking | Yes | 0.032 | **2.03** | 1.06 | 3.86 |
| | No | | | | |

For multiple categories, the final one is the reference. Odds ratios with significant *p*-values are in bold.

The multivariable logistic regression analysis predicted higher psychedelic use among medical students without intact religious beliefs or commitment to religious practice (OR = 7.26, 95% CI [2.33, 22.60], *p* = 0.001). Students who perceived sporadic psychedelic use as low-risk or without risk (3.03, [1.44, 6.36], *p* = 0.003 and 3.01, [1.37, 6.63], *p* = 0.006, respectively) were more prone to engage in using psychedelics, with a notable concentration of this trend among Poles. Further, medical students residing in Poland (2.82, [0.83, 9.55], *p* = 0.095), those holding positive views on psychedelics (2.49, [1.20, 5.14], *p* = 0.014), and those frequently (daily) venturing outside their home during the evening (2.41, [1.26, 4.61], *p* = 0.008) also demonstrated an increased propensity toward psychedelic use. Moreover, male students (2.05, [1.10, 3.85], *p* = 0.025) and those who smoked cigarettes or tobacco (2.03, [1.06, 3.86], *p* = 0.032) were likelier to engage in psychedelic use.

Tables 2 and 3 represent a multivariable analysis using a binary logistic regression model incorporating multiple parameters (independent variables). Unlike classical univariable statistical testing, a significant parameter at a 90% confidence interval may hold importance when considering a larger sample or additional variables. Besides, the Chi-squared test revealed a significant difference in psychedelic use between the Iraqis and Poles. On another note, the odds ratio, indicating the effect size, was smaller in the multivariable test compared to the univariable analysis.

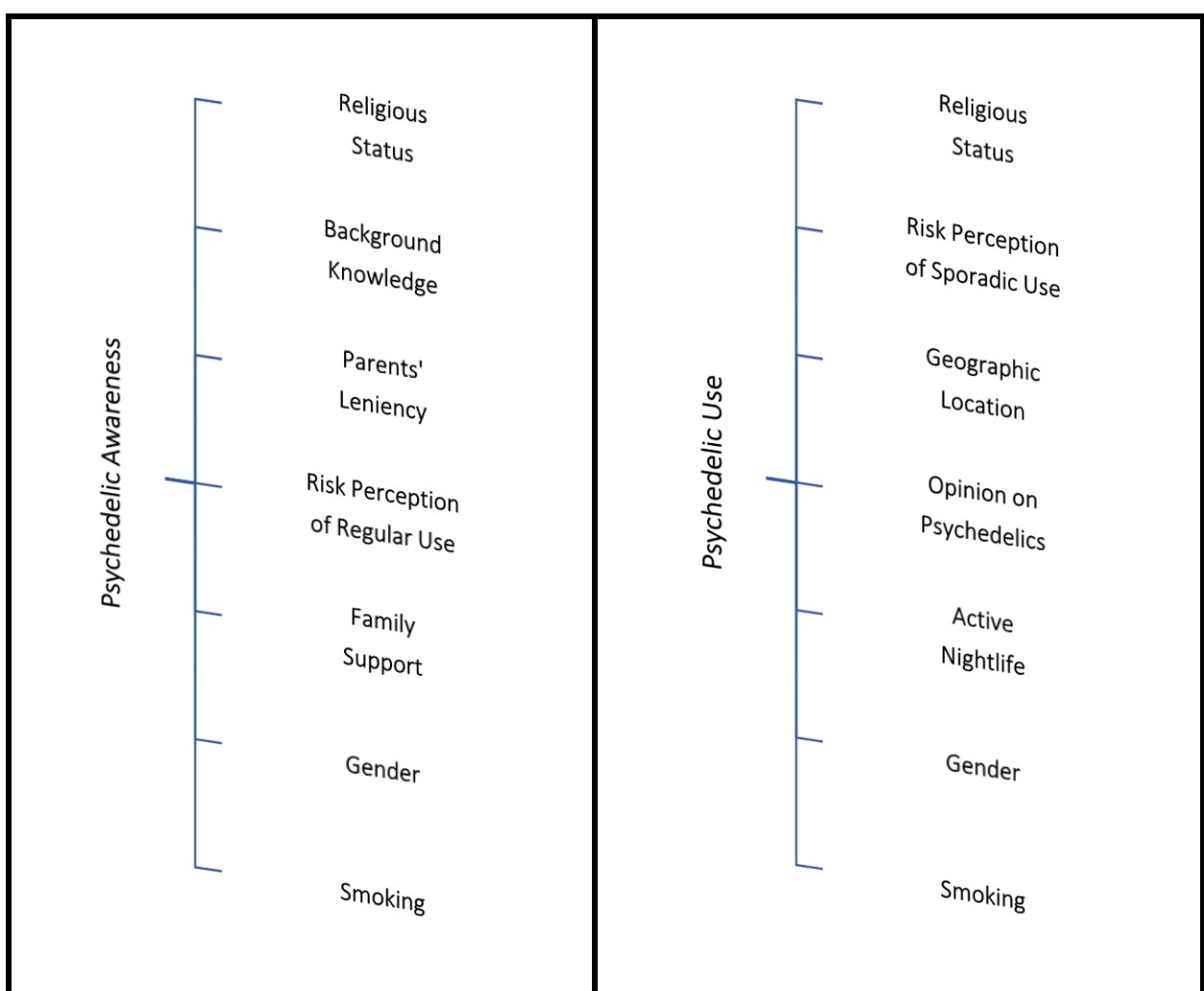

**Figure 3.** Factors influencing knowledge of psychedelics and their use.

To conclude, the most critical risk factors, as indicated by their odds ratios, point to a greater likelihood of psychedelic use among medical students who lack an affiliation with any particular religious belief or a commitment to religious practice, irrespective of the specific nature of the religious belief system. Similarly, students who perceive sporadic psychedelic use as low-risk (or without risk) and those residing in Poland also demonstrate an elevated propensity toward such usage. These findings align with our initial study hypothesis and corroborate earlier observations regarding the socio-demographic and socio-cultural differences between Polish and Iraqi medical students.

## 4. Discussion

It should be emphasized again that the countries from which the study participants were recruited are culturally diverse, which is very likely to affect the results obtained. Significant disparities in the utilization and awareness of psychedelics were observed among the Polish and Iraqi medical students. The Polish medical students displayed a notably higher rate of psychedelic usage than their Iraqi counterparts, highlighting a stark contrast in familiarity with these substances. All five Iraqi medical students who used psychedelics admitted to LSD use, constituting 100% of the group. However, it is unclear whether they obtained the substance locally or abroad while traveling. Conversely, the Polish medical students exhibited greater diversity in psychedelic use, with LSD (51 students; 75%), ayahuasca (39; 57.4%), psilocybin (37; 54.4%), NBOMe (31; 45.6%),

DMT (13; 19.1%), ketamine (5; 7.4%), and 5-MeO-DMT (3; 4.4%) being the most commonly used substances.

Several key factors contributed to higher psychedelic use, including a reduced commitment to religious beliefs, positive attitudes toward psychedelics, and social nightlife activities. Additionally, being male and tobacco use were associated with increased psychedelic use. These findings underscore the importance of recognizing and addressing these variations using culture- or population-specific management strategies.

Taking into account religiosity, which was one of the main variables (highest effect size) differentiating the two groups in our study, significant differences were observed. Among the Polish respondents, significantly more participants emphasized their casual or indifferent attitude toward religion and the norms associated with its practice. At the same time, in this group, a higher percentage confirmed the use of psychedelics and other mood-altering drugs. The opposite was true among Iraqis, who were much more likely to declare a religious affiliation and, at the same time, had a significantly lower interest in using psychedelics.

Any normative attitudes associated with religiosity can influence the use of any mood-altering drugs, including psychedelics. Many studies suggest that people who regularly practice their faith or consider religion an essential part of their lives are less likely to experiment with drugs [17–19]. Dalgalarrondo and colleagues have conducted such studies among Brazilian adolescents (*n* = 2287) [20], Marsiglia and team among American adolescents (*n* = 7304) [21], and Engs and Mullen among Scottish students (*n* = 4065) [22], among others. Also, Polish studies comparing religiosity with attitudes toward psychoactive drugs (*n* = 2273) [23] had corresponding results to those reported in the current study. In each of the measurements conducted, the researchers found that indications of religious affiliation and participation in religious practices were identified as determinants that were strongly protective against drug use.

Religiosity can influence the use of psychedelics, which is particularly relevant in Muslim countries, where the consumption of mind-altering substances (including alcohol) is often strictly prohibited for religious reasons. Religious teachings and cultural norms in Muslim countries are protective against drug use and possession [24]. In Polish society, however, as Janusz Marinski notes, religiosity takes on different faces: alternative, multidimensional, non-dogmatic, non-institutionalized, dynamic, fluid, loose, scattered, syncretic, pluralistic, and individualized. They develop according to the needs and desires of citizen consumers. Thus, one can speak of increasing freedom in the choices made [25], including whether to use mood-altering agents.

A study by Viner et al. found that Muslims are far more strongly condemning of the use of psychoactive substances and are less likely to drink alcohol than followers of other religions [26]. However, as seen in our study, there were also respondents in Iraq choosing to use such drugs. Nonetheless, this is not a surprising phenomenon in the country. Many years of political, economic, and social instability are conducive to the search for both sources of income and, above all, solace and respite from the psychological tensions experienced for decades in Iraq [27,28].

It is essential to recognize the potential for a transformative shift in mental health treatment, especially within the Iraqi community, through the therapeutic use of psychedelics. This community has endured significant trauma due to sanctions in the 1990s and the conflict with the Islamic Republic of Iraq in the 1980s. As the therapeutic potential of psychedelics gains popularity, more individuals are exploring their use independently [29]. The pursuit of medical education among our respondents is likely to foster a deeper understanding and interest in using psychedelics for therapeutic purposes, potentially bypassing established societal norms.

Studies also suggest that psychedelics can be used as tools to transmit beliefs. Many indigenous groups use psychedelics in communal rituals, suggesting that these substances are potent catalysts for social affiliation, enculturation, and belief transmission [30]. Whether this occurs in the case of the Iraqi community is an unrecognized issue, which is why re-

search on drug use among Muslim populations is essential to understanding and effectively dealing with this problem [31].

An alternative attempt to explain and understand the differences in psychedelic use between Iraqis and Poles may be that cultural differences may originate from conditions other than religiosity and religious observance. It is possible that, along with religious differences, there are different mechanisms of external control (laws, rules) affecting respect for these rules (internal control). Iraqi society has been considered a society with solid traditions and control mechanisms internalized by Iraqis and strictly observed [32]. Of course, these mechanisms exist in every community and are designed to integrate society to protect against undesirable behavior, promoting security for all citizens.

Our research shows that Polish respondents have much more casual relationships with relatives, devoid of closeness and attention, which corresponds very well with the cited theories. The attachment to family members and strong emotional ties between parents and children observed among Iraqis foster adherence to accepted norms and may effectively deter them from engaging in any behavior stigmatized in that culture, including using psychedelics. Rules based on religious norms and strong family ties cannot be easily relativized.

It is also essential to have a diverse knowledge of psychedelics and, simultaneously, of the availability of these drugs among the populations (nations) studied. Worldwide, there has been a renaissance of research on psychedelics in recent years. In Poland, the Polish Psychedelic Society was founded in 2019 as part of the activities of the Polish Drug Policy Network to popularize reliable knowledge about psychedelics, integrate the experience of experts from various fields, and work for the rational regulation of psychedelic substances. The society has many goals, including the creation of an interdisciplinary platform for cooperation among professionals working on the topic of psychedelics; the promotion of scientific research on psychedelics, with a particular focus on their therapeutic potential; and the education of the public by providing factual information based on scientific research on the positive and negative consequences associated with the recreational and therapeutic use of psychedelics.

All its information and materials are available on the society's website [33]. The internet, moreover, has become a vital source of information on psychedelics. For example, streaming platforms such as Netflix and Amazon Prime offer movies and documentaries on the subject. Websites with testimonials from users who use psychedelics recreationally are also an indispensable source of information [34]. Polish students, who spend much more time browsing websites, certainly have many more opportunities to acquire expertise in this area than their Iraqi counterparts, who report closer ties with relatives, religious observance, and less time spent on recreational internet use. Since 2003, Iraqi internet users have had unrestricted access to the web without censorship of political, global news, religious, or cultural content. However, measures are in place to filter pornography websites aimed at protecting minors under 18 years of age.

An exciting issue highlighted in our research is the self-induced microdosing of psychedelics, which some respondents admitted. Microdosing on psychedelics, regularly taking minimal (sub-hallucinogenic) doses of a psychedelic substance such as LSD or psilocybin-containing mushrooms, has gained popularity in recent years. On another note, people who microdose with psychedelics report minimal acute effects of these substances but say the long-term health and well-being benefits are significant [35,36]. Furthermore, over the past ten years, microdosing research has expanded its scope from the basic science of its molecular mechanisms to vaccines for prevention in high-risk groups, drug development, psychotherapy, innovative treatment delivery, and the integration of substance abuse treatment into the overall medical system [35].

The Psychedelic Experiences Survey, an international online survey examining the practices and subjective experiences of using psychedelic substances in standard doses and microdoses, conducted in late 2018, gathered insightful results. Diverse motivations for using psychedelics—psilocybin, LSD and 1P-LSD (an LSD analog), and N,N-

dimethyltryptamine (DMT)—were identified among 525 people reporting microdosing. The study found that more than half of the subjects had taken a standard (regular) dose of psychedelics before and only later began using microdoses, with varying justifications. The most common primary motivations were solving mental health or substance use problems (40.4%), personal and spiritual growth (31.2%), and improving cognitive performance at work or in studies (18.1%). It should be noted that more than half of the sample (53%) reported higher education, and 17% described themselves as currently studying. Therefore, it can be presumed that both the acquisition of knowledge in this area and the practice of using microdoses of psychedelics were a consequence of higher education, through which they were able to learn such knowledge (from friends, the internet, and curricula, among other things) [37].

A 2023 publication reviewing 42 studies on psilocybin use found that the results of psilocybin-microdosing studies suggest enhanced cognition, productivity, and creativity. Moreover, the microdosing studies, including a long-term follow-up, showed its neutral and positive effects on cognitive performance. However, the authors did not believe meaningful conclusions could be drawn from these data [38].

Our research may point to the beginning of this process in Iraq, but without a corresponding study completed on a much larger sample or aimed at focusing on this area, it is difficult to determine this. National laws in many countries apply strict policies in response to drug use and possession. These legal responses could be based on cultural or religious doctrine that prohibits the consumption of mind-altering substances, as we have written about previously. Qualitative research may be needed to determine the risk factors because of the stigma associated with substance use in Muslim-majority nations [35]. Such a phenomenon is intriguing and is worthy of separate observation and measurement among Poles and Iraqis.

Psychedelics represent a topic of global scientific inquiry, with the research extending its reach to Poland. While quantifying the exact number of Polish individuals who engage in psychedelic use proves challenging, ongoing investigations shed light on emerging patterns. A 2019 ESPAD study encompassing nearly 6000 school-aged participants in Poland found that the use of psychedelics, primarily LSD and other hallucinogens excluding mushrooms, ranged from 3.2% to 4.2%, varying by age group. Approximately 2% of respondents openly acknowledged their consumption of hallucinogenic mushrooms. Notably, psychedelics held a moderate appeal among young individuals seeking altered states of consciousness. Their preferences leaned more toward substances like marijuana, hashish, over-the-counter drugs, or combinations of alcohol with either marijuana or other drugs. The prevalence of adolescent psychedelic usage has remained relatively consistent over the past two decades, encompassing the span of this survey (ranging from 2.7% to 5.2%), with the 2019 survey observing a decline in interest in these substances [39].

In early 2020, an online survey encompassing 443 participants unveiled that the primary impetus behind psychedelic use leaned toward personal growth rather than social or recreational objectives [40]. Conversely, in 2021, a comprehensive study on young Polish individuals sought to identify harm reduction strategies employed while self-medicating with various psychoactive substances for mental health challenges. This study boasted a substantial sample size, comprising thousands of respondents. Its findings indicated that when it came to self-medicating for conditions such as depression, anxiety disorders, substance abuse disorders, borderline personality disorder, and bipolar affective disorder, psychedelics emerged as the most efficacious agents. Remarkably, the success rate of self-medication with psychedelics was nearly 80% [41].

In 2021, one of the most comprehensive investigations into Polish psychedelic users was undertaken, yielding notable insights. An online survey collected data from an impressive cohort of 2516 participants, among whom 1661 (accounting for 66%) confirmed their use of psychedelics. These substances included LSD, psilocybin mushrooms, synthetic psilocybin, DMT, "Changa" (a smoking mixture comprising DMT and β-carbolines), ayahuasca, mescaline, and other derivatives of classic psychedelics. The study's findings

revealed that frequent engagement with psychedelics within a naturalistic setting was correlated with heightened positive emotional responses and the development of a constructive self-awareness framework. The psychological effects associated with psychedelics demonstrated a positive and potentially beneficial impact on individuals' psychological well-being [42].

In late 2022, a study involving 1117 respondents highlighted an upward trajectory in using hallucinogenic mushrooms and DMT within Poland. This trend has prompted calls from the survey's authors to prioritize public education as a critical component of public health initiatives [43]. The former suggests that the fascination with psychedelics has been a discernible phenomenon among young Poles for at least two decades, increasingly adopted for self-medication and enhanced well-being. Notably, this trend is not exclusive to Poland; it is a broader trend across Europe. A significant corroboration comes from the 2023 report by the European Monitoring Centre for Drugs and Drug Addiction (EMCDDA), which underscores the burgeoning interest across European nations in the clinical and therapeutic applications of various psychedelics, including lesser-known substances, to addressing mental health disorders. However, the report also underscores the importance of caution, emphasizing that despite the promising research outcomes, self-medication or experimentation with these substances without proper medical guidance carries inherent risks [44].

*Study Limitations*

The present study does exhibit certain limitations, primarily stemming from the relatively imbalanced ratio (1.35:1) between the Iraqi and Polish students who completed the survey. This disproportionate representation can be attributed to two key factors: (1) the distribution of the survey across multiple medical colleges and universities in Baghdad (the University of Baghdad, Al-Nahrain University, Al-Iraqi University, and Al-Mustansiriyah University), as opposed to the solitary medical university in Poznan (Poznan University of Medical Sciences) and (2) the higher overall population density of Baghdad when contrasted with Poznan.

Further limitations arise from the inherent constraints of the cross-sectional study design. Specifically, the inability to establish the direction of causality between an exposure (cause) and an outcome (effect) is attributed to the absence of a comprehensive longitudinal perspective, which precludes a nuanced examination of the temporal arrow of time (temporality). However, it is essential to underscore that assigning variables as independent and dependent was not arbitrary. Our study adhered to the well-established Bradford Hill causality assessment criteria, ensuring a rigorous and principled approach to causal inference.

Another limitation of cross-sectional surveys arises from their anonymity component. Anonymity in survey responses encourages honest participation and protects respondent privacy, but it hampers identity verification, potentially affecting data accuracy and generalizability, particularly in online surveys. Researchers must remain vigilant to mitigate biases and inaccuracies in the collected data. Additionally, we acknowledge the possibility that some respondents, whether using or abstaining from psychedelics, may have preexisting psychopathologies or psychiatric comorbidities. However, screening or excluding such students would pose challenges and hinder the survey response rates.

Concerning age, the ESPAD project primarily focuses on 16-year-old students. In contrast, our ESPAD-adapted survey was intentionally designed to encompass a broader demographic range, explicitly targeting junior students (those at or below 20 years of age) and senior students (those aged 20 years and older) from college and university settings. We must acknowledge another notable limitation of our study: the limited participation of individuals who reported using psychedelics to enhance their academic performance. Ten students (one Iraqi and nine Polish participants) contributed to this category in this context. Due to the small sample size in this specific subgroup, this parameter (cognitive enhancement) was excluded from the multivariable analysis of psychedelic use.

Moreover, the multivariable analysis exhibited a relatively minor limitation in terms of sensitivity, an aspect that could be significantly enhanced by expanding our sample size and including additional relevant parameters. Importantly, it is worth noting that the overall statistical power and predictive values remain robust and indicative of the soundness of our findings. Another crucial issue relates to the availability of psychedelics in Iraq and Poland, and it was challenging to incorporate this factor into our regression model due to its complexity and limited data. However, the geographic location (Poland and Iraq) is a proxy representation of the availability of psychedelic substances. On another note, the availability of substances like amphetamines is well documented (most prevalent) on the illicit drug market in Iraq, whereas psychedelics are much less accessible.

Furthermore, regarding the awareness of psychedelics, erroneous positive responses are possible. This phenomenon arises from flawed awareness (pseudo-awareness) of psychedelics. Notably, instances of such pseudo-awareness were more conspicuous among the Iraqi participants. Pseudo-awareness may also manifest if participants resort to guessing or cheating by searching for correct answers on external resources, including the internet. However, it is crucial to emphasize that these pseudo-aware participants were identified (15 Iraqi participants) and excluded from the statistical analysis. Another potential source of bias could emerge in the computation of the psychedelic awareness score. Nonetheless, the criteria for calculating these scores were agreed upon before distributing the survey, as exhaustively outlined in the Methods section. Additionally, the scoring process was conducted by two independent assessors (raters) unaware of each other's assessments (double-masked approach). When discrepancies in score calculation materialized, a third rater was involved in reconciling any disparities between the initial two raters.

In advancing psychedelic epidemiological research, a promising opportunity lies in integrating robust causality assessment methods [45] and implementing artificial intelligence (AI) techniques to facilitate comprehensive analysis of large datasets [46,47]. Additionally, substantial methodological refinement can be achieved by incorporating rigorous systematic reviews and meta-analyses, enhancing the quality and depth of the evidence regarding the awareness and utilization patterns for psychedelic substances within specific target populations.

## 5. Conclusions

Our study highlights significant disparities in psychedelic use and awareness among participants from Iraq and Poland. Notably, cultural context plays a pivotal role in shaping these patterns. The key risk factors for higher use of psychedelics include non-commitment to a religious belief or its practice, perceiving psychedelic use as low- or no-risk, geographic location, a favorable opinion of psychedelics, lively (active) outdoor social nightlife activities, male gender, and smoking. The Polish participants exhibited an almost 20-fold higher prevalence of psychedelic use and a heightened level of awareness, whereas the Iraqis tended to perceive psychedelics as having a higher level of risk. The former findings underscore the need for context-specific interventions. In Iraq, where psychedelic use is less prevalent, public health efforts could focus on increasing awareness of the associated risks. In Poland, where use is more prevalent, targeted interventions can address safe usage practices in conjunction with risk perception.

**Supplementary Materials:** The following supporting information can be downloaded at: https://www.mdpi.com/article/10.3390/ejihpe14030038/s1, Table S1: The English version of the ESPAD-adapted survey; Table S2: Socio-demographic and socio-cultural differences between Poles and Iraqis.

**Author Contributions:** Conceptualization, A.A.-I. and M.M.; data curation, A.A.-I. and M.M.; formal analysis, A.A.-I., M.A.M. and M.M.; funding acquisition, A.A.-I., M.A.M. and M.M.; investigation, A.A.-I., M.A.M. and M.M.; methodology, A.A.-I., M.A.M. and M.M.; project administration, A.A.-I.; resources, A.A.-I. and M.A.M.; software, A.A.-I. and M.M.; supervision, M.M.; validation, A.A.-I. and M.M.; visualization, A.A.-I.; writing—original draft, A.A.-I. and M.A.M.; writing—review and editing, A.A.-I., M.A.M. and M.M. All authors have read and agreed to the published version of the manuscript.

**Funding:** The current research was financed by a statutory subsidy for young scientists at the Doctoral School of Poznan University of Medical Sciences. The decision by the Grant Evaluation Committee was issued as per the document numbered SDUM-DGB 14/05/22 on 5 May 2022.

**Institutional Review Board Statement:** The study followed the ethical guidelines outlined in the Declaration of Helsinki by the World Medical Association and the Framingham Consensus of 1997. While no approval from the bioethics committee was necessary, it underwent examination by the Institutional Review Board (Ethics Committee) of Poznan University of Medical Sciences (Uniwersytet Medyczny im. Karola Marcinkowskiego w Poznaniu). It was exempted from approval requirements on 12 November 2021 due to its observational (non-experimental) nature.

**Informed Consent Statement:** Informed consent was obtained from each study participant.

**Data Availability Statement:** Data are available upon request from the corresponding author.

**Acknowledgments:** The authors express their sincere gratitude for the dedicated contributions of the following undergraduate medical students: Albab Ahmed Hekmat, Ali Hussein Hameed, Fatima Laith Mohammed, Tabarak Rafid Abdul-Rahman, Zahraa Saad Abdul-Munim, and Zeena Mahmod Nabeeh. Their invaluable efforts extended beyond distributing the survey via the university's mailing list, encompassing the dissemination of the survey in print form to their esteemed colleagues from the College of Medicine at the University of Baghdad. These students significantly enhanced the outreach and impact of this research. Ahmed Al-Imam participates in the STER Internationalisation of Doctoral Schools Programme from NAWA, the Polish National Agency for Academic Exchange No. PPI/STE/2020/1/00014/DEC/02.

**Conflicts of Interest:** The authors declare no conflicts of interest.

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
