# Peer review of "Cultural Divergence in Psychedelic Use among Medical Students: An ESPAD-Adapted Survey among Poles and Iraqis"

_ejihpe, doi:10.3390/ejihpe14030038_

Round 1

Reviewer 1 Report

Comments and Suggestions for Authors

The study presented by Al-Imam and colleagues affords a very interesting perspective of how psychedelic use might be altered in function of the culture, and more concretely between Europe and Asia countries. Despite this nice goal, the manuscript offerings a lot of deficiencies, and in somehow it seems it has been edited by copy and paste from a previous revision elsewhere. Nonetheless is authors improve the quality of the document and data presentation, it would be a nice work to be published.

Introduction:

- It is too long. The part of the gulf war, is it necessary? I mean, these students that made the survey were not born yet..

- Which OTC can be considered as a drug abuse?

- Lines 156-162 deal with adolescent females. There is nothing in the discussion regarding to this, focusing on it.

Material and Methods

- Here it should be written the number of people who received the survey and did not answer.

- Sample size and Universities involved should appear here firstly.

- Statistical analysis and methods do not correlate with the analysis shown such as Chi squared test, Mann-Whitney U test and so on…

- Line 181 please change focus of for focus on

Results

- Data provide in lines 246-251 are contradictory with the Figure 1 (I did not find it in the text). Line 269 names Figure 1 but it is not related to the one of page 7 I would say.

- Line 276 “Iraqis tended to miss university classes deliberately”, why?

- Lines 292-297: students are not familiar with psychedelic drugs, but in the introduction it is said the use it is increasing from 2003. Clearly authors are referring to two different population ages and not the aim of the study.

- Figure 3 is not clear, is there any statistical analysis or not?

- Table 1 compares Polish students with Iraq ones, right?

- Table 1: 0.057 is significant? It is in bold! Please add in the footnote the analysis performed.

- Table 2: all Odds Ration are in bold, how can be Agree 1.17 and Disagree 1.91 both significant? P-value of 0.065 is significant? Please explain this analysis.

- Table 3: please explain as in Table 2 all Odd Ratio are significant (line 420 p=0.095)

Discussion

- It is also too long and the message of the paper is diluted. Please re-write it more focused on the results provided in the paper.

- For instance along the introduction it is said how a psychedelic acts. However, nothing is shown regarding which one was the most used in the two population of study.

- Another point that I do not see,  Iraq students have free access to internet or is there any censure?

Comments on the Quality of English Language

Minor editing of English language required

Author Response

Prof. Dr. José Jesús Gázquez Linares,

European Journal of Investigation in Health, Psychology and Education (EJIHPE),

Almería, Spain.

Poznan – Poland, 11th of February 2024

Subject: Revised Manuscript

Dear Prof. Dr. José Jesús Gázquez Linares, Editor-in-Chief of the EJIHPE Journal,

Dear Prof. Dr. María del Carmen Pérez-Fuentes, Editor-in-Chief of the EJIHPE Journal,

Dear Ms. Esther Liu, Managing Editor of the EJIHPE Journal,

             On behalf of all co-authors, we affirm our strict adherence to the valuable feedback the three esteemed reviewers provided. The revised version has garnered unanimous approval from all co-authors. Utilizing the track-changes option in Office Word, we edited the manuscript. We sincerely appreciate the review process's thoroughness and thank the journal's editors and reviewers for their dedicated efforts. Please find our responses to each comment below.

Reviewer-1

  • The study presented by Al-Imam and colleagues affords a very interesting perspective of how psychedelic use might be altered in function of the culture, and more concretely between European and Asia countries. Despite this nice goal, the manuscript offers many deficiencies, and somehow, it seems it has been edited by copy and paste from a previous revision elsewhere. Nonetheless, if authors improve the document's quality and data presentation, it would be nice work to publish.

Response: Dear Sir/Madam, thank you for your time and efforts in reviewing our manuscript. We appreciate your positive feedback on the significance and novelty of our study. We confirm our commitment to addressing the limitations and deficiencies you highlighted. We also confirm that this study underwent peer review for the first time in your esteemed journal, and this is our initial revision. However, we acknowledge that we extensively edited the manuscript before and after engaging with a few journals' editors to ensure a favorable editorial response (welcoming the submission) before passing to the peer review stage.

  • It is too long. The part of the Gulf War, is it necessary? I mean, these students that made the survey were not born yet.

Response: Dear Sir/Madam, thank you for your remark. We've trimmed the introduction as much as possible. Regarding your point about the Gulf War, it occurred in 1990-1991, distinct from Saddam Hussein's government's fall in 2003. While we recognize the importance of these details, we've condensed the paragraph accordingly. On another note, the ripple effects of the American invasion of Iraq in 2003 extend beyond the immediate generation to influence subsequent younger generations, transmitting a legacy of skepticism towards government actions, heightened awareness of global issues, and potential impacts on mental health, economic opportunities, political engagement, and perceptions of security. This intergenerational transmission further underscores the profound and enduring consequences of the conflict on the youth who came of age in its aftermath.

  • Which OTC can be considered as a drug abuse?

Response: Dear Sir/Madam, thank you for your query. Certain over-the-counter (OTC) drugs, like cough and cold medications with ingredients such as dextromethorphan (DXM) or codeine, antihistamines like diphenhydramine, pain relievers such as ibuprofen or acetaminophen, laxatives, and caffeine pills, can be abused for their sedative effects, pain relief, weight loss, or performance enhancement. Additionally, some of these OTCs possess psychoactive or neurotropic properties, including codeine, DXM, and antihistamines.

  • Lines 156-162 deal with adolescent females. There is nothing in the discussion regarding to this, focusing on it.

Response: Dear Sir/Madam, thank you for your insightful feedback. We've significantly shortened the paragraph while integrating it with pertinent revisions and edits in the discussion section. On another note, as written in the abstract, most respondents in the current study were females (65.6%).

  • Material and Methods. Here it should be written the number of people who received the survey and did not answer.

Response: Dear Sir/Madam, thank you for your insightful remark. We have revised the "Sampling and data collection" subsection of the methods. The survey was distributed to 3,768 students, comprising 1,306 Poles and 2,462 Iraqis. However, only 739 students responded and completed the survey, with 315 respondents from Poland and 424 from Iraq.

  • The sample size and Universities involved should appear here first.

Response: Dear Sir/Madam, thank you for your keen observation. Based on your feedback, we relocated the "Research ethics" subsection to the end of the methods section and revised the "Sampling and data collection" subsection.

  • Statistical analysis and methods do not correlate with the analysis shown, such as the Chi-squared test, Mann-Whitney U test, and so on.

Response: Dear Sir/Madam, thank you for your keen observation. We confirm the use of binary logistic regression, Chi-squared test, and Mann-Whitney U test in our analyses, progressing from univariable to multivariable analyses. Additionally, Prof. Michal Michalak, an expert in medical biostatistics and data science, supervised the statistical analyses as our research team leader.

  • Line 181, please change "focus of" for "focus on"

Response: We have revised it per your remarks. Thank you.

  • Data provided in lines 246-251 are contradictory with Figure 1 (I did not find it in the text). Line 269 names Figure 1, but it is not related to the one on page 7, I would say.

Response: Dear Sir/Madam, thank you for your query. Figure 1 displays the frequencies of "psychedelic use" (top) and the "Whereabouts Awareness During Weekend Nights" (bottom). These charts represent the default output of the statistical package (IBM-SPSS) for the Chi-squared test examining the association between nationality and psychedelic use and between nationality and whereabouts awareness. The observed frequencies are conveyed in Figure 1. However, to eliminate confusion, we have removed the top graph, as the difference in psychedelic use among Iraqis and Poles was previously discussed in the subsection "The prevalence of psychedelic use in Iraq and Poland.".

  • Line 276: "Iraqis tended to miss university classes deliberately." why?

Response: Dear Sir/Madam, thank you for your question. Following the 2003 invasion of Iraq, various factors could have contributed to Iraqis deliberately missing school attendance. These include security concerns stemming from the conflict, displacement, and infrastructure damage disrupting schooling, economic hardship hindering families' ability to afford education-related expenses, psychological trauma from the violence experienced, and cultural or social factors such as gender discrimination.

  • Lines 292-297: students are not familiar with psychedelic drugs, but in the introduction, it is said the use it has been increasing from 2003. Clearly authors are referring to two different population ages and not the aim of the study.

Response: Dear Sir/Madam, thank you for your keen observation. In the introduction, it is highlighted that in Iraq, there has been an increase in the abuse of certain substances, including psychostimulants, cannabis, opioids, and over-the-counter drugs (OTC), especially post-2003. While some of these substances are psychoactive or novel psychoactive substances (NPS), they do not fall under the category of psychedelic substances. In our study, Iraqi medical students exhibited notably lower awareness of psychedelic drugs compared to their knowledge of other psychoactive substances, some of which could be considered quasi-psychedelics, like amphetamines and amphetamine-type stimulants (ATS). However, it's noted that these substances have distinct mechanisms of action and pharmacodynamics from psychedelics. Additionally, there is a lack of large-scale surveillance or epidemiological mapping regarding awareness and use of psychedelics among Iraqis, particularly among the medical student population.

  • Figure 3 is not clear. Is there any statistical analysis or not?

Response: Dear Sir/Madam, thank you for your query. All graphs in Figure 3 correspond to the statistical analysis outlined in the relevant paragraph and Table 1. In essence, Table 1 and Figure 3 are interconnected, with Figure 3 depicting a subset of four parameters of interest extracted from Table 1. These parameters include "Using the Internet for Leisure," "Risk of Sporadic Use of Psychedelics," "Parents' Strict Rules Outside Home," and "Opinions on Psychedelics." To avoid confusion, we deleted Figure 3.

  • Table 1 compares Polish students with Iraq ones, right?

Response: Dear Sir/Madam, Thank you for your question. Table 1 pertains to the entire (total) sample and provides a comparison of knowledge about psychedelics across various survey parameters, such as age, religious status, etc., including opinions on psychedelics. The differences between Polish and Iraqi medical students are elaborated in detail in the "Socio-demographic and socio-cultural differences" of the results.

  • Table 1: 0.057 is significant? It is in bold! Please add the analysis performed in the footnote.

Response: Dear Sir/Madam, thank you for your keen observation. We sincerely apologize for the error. A p-value of 0.057 is not significant at a 95% confidence interval. Table 1 has been corrected accordingly, and the footnote has been revised to specify the statistical analysis performed: "The p-value relates to the total score of psychedelic awareness. Significant p-values are in bold fonts. The statistical analysis was based on non-parametric univariable models: Kruskal-Wallis H and Mann–Whitney U tests.".

  • Table 2: All odds ratios are in bold, how can be "Agree" (1.17) and "Disagree" (1.91) both significant? P-value of 0.065 is significant? Please explain this analysis.

Response: Dear Sir/Madam, Thank you again for your astute observation. We removed the bold font from the "agree" category as its p-value was insignificant. The interpretation suggests that students who cannot (the "disagree" category) discuss their problems with their families possessed higher awareness concerning psychedelics, in contrast to their peers who can freely communicate with their parents or families about their problems.

  • Table 3: Please explain as in Table 2, all Odd Ratio are significant (line 420 p=0.095)

Response: Dear Sir/Madam, thank you for your comment. Table 3 represents a multivariable analysis using a binary logistic regression model incorporating multiple parameters (independent variables). Unlike classical univariable statistical testing, a significant parameter at a 90% confidence interval may hold importance when considering a larger sample or additional variables. Additionally, the Chi-squared test revealed a significant difference in psychedelic use between Iraqis and Poles. On another note, the odds ratio, indicating effect size, was smaller in the multivariable test compared to the univariable analysis.

  • It is also too long, and the message of the paper is diluted. Please re-write it more focused on the results provided in the paper.

Response: Dear Sir/Madam, thank you for your request. We shortened and refined the discussion section by removing three paragraphs related to Hirsch and LeBlanc's theory. Additionally, we incorporated new data in the first paragraph regarding the most commonly used psychedelics among Polish and Iraqi medical students. Furthermore, we updated the study limitations subsection according to feedback from other reviewers.

  • For instance, along the introduction, it is said how a psychedelic acts. However, nothing is shown regarding which one was the most used in the two populations of the study.

Response: Dear Sir/Madam, thank you for your crucial query. We added those details to the first paragraph of the discussion section: "All five Iraqi medical students admitted to LSD use, constituting 100% of the group. However, it is unclear whether they obtained the substance locally or abroad while traveling. Conversely, Polish medical students exhibited greater diversity in psychedelic use, with LSD (51 students; 75%), ayahuasca (39; 57.4%), psilocybin (37; 54.4%), NBOMe (31; 45.6%), DMT (13; 19.1%), ketamine (5; 7.4%), and 5-MeO-DMT (3; 4.4%) being the most commonly used substances.".

  • Another point that I do not see is that Iraq students have free internet access or if there is any censure.

Response: Since 2003, Iraqi internet users have had unrestricted access to the web without censorship on political, global news, religious, or cultural content. However, measures are in place to filter pornography websites aimed at protecting minors under 18 years of age.

Once more, we sincerely thank the respected reviewers for their dedicated efforts in reviewing our manuscript. We remain open to addressing additional concerns regarding our manuscript and are fully committed to meeting all specified requirements. Thank you for your continued consideration and support.

Best regards,

The corresponding author.

Reviewer 2 Report

Comments and Suggestions for Authors

 The topic of the research is certainly interesting and gives an idea of how medical students behave towards psychedelic drugs in these two nations. However, there are some important limitations:

-The questionnaireEuropean School Survey Project on Alcohol and 17 Other Drugs (ESPAD),  was sent via email and, despite the measures used, we have no idea who actually filled out the questionnaire

- There are obvious issues relating to face validity

- we have no idea of the real actual psychic balance of those who responded to the test.

These could be suggestions for an implementation of the research in the future and not just criticisms of the current version of the work carried out.

Author Response

Prof. Dr. José Jesús Gázquez Linares,

European Journal of Investigation in Health, Psychology and Education (EJIHPE),

Almería, Spain.

Poznan – Poland, 11th of February 2024

Subject: Revised Manuscript

Dear Prof. Dr. José Jesús Gázquez Linares, Editor-in-Chief of the EJIHPE Journal,

Dear Prof. Dr. María del Carmen Pérez-Fuentes, Editor-in-Chief of the EJIHPE Journal,

Dear Ms. Esther Liu, Managing Editor of the EJIHPE Journal,

             On behalf of all co-authors, we affirm our strict adherence to the valuable feedback the three esteemed reviewers provided. The revised version has garnered unanimous approval from all co-authors. Utilizing the track-changes option in Office Word, we edited the manuscript. We sincerely appreciate the review process's thoroughness and thank the journal's editors and reviewers for their dedicated efforts. Please find our responses to each comment below.

Reviewer-2

  • The topic of the research is certainly interesting and gives an idea of how medical students behave towards psychedelic drugs in these two nations. However, there are some important limitations:

Response: Dear Sir/Madam, Thank you for your opinion and kind words regarding the importance of the current study. We also confirm compliance by addressing the limitations you highlighted in your report.

  • The questionnaire European School Survey Project on Alcohol and 17 Other Drugs (ESPAD) was sent via email, and, despite the measures used, we have no idea who actually filled out the questionnaire.

Response: Dear Sir/Madam, thank you for your keen and insightful remarks. We agree with them. We have revised the limitations subsection of the discussion to address your observation. Anonymity in survey responses encourages honest participation and protects respondent privacy, but it hampers identity verification, potentially affecting data accuracy and generalizability, particularly in online surveys. Researchers must remain vigilant to mitigate biases and inaccuracies in the collected data.

  • There are obvious issues relating to face validity.

Response: Dear Sir/Madam, thank you for highlighting this critical issue regarding the validity and reliability of the results. We have addressed this concern in the limitations subsection of the results: "Concerning age, the ESPAD project primarily focuses on 16-year-old students. In contrast, our ESPAD-adapted survey was intentionally designed to encompass a broader demographic range, explicitly targeting junior students (those at or below 20 years of age) and senior students (those aged 20 years and older) from college and university settings. We must acknowledge another notable limitation in our study: the limited participation of individuals who reported using psychedelics to enhance their academic performance. Ten students (one Iraqi and nine Polish participants) contributed to this category in this context. Due to the small sample size in this specific sub-group, this parameter (cognitive enhancement) was excluded from the multivariable regression analysis of psychedelic use.".

  • We have no idea of the actual psychic balance of those who responded to the test.

Response: Dear Sir/Madam, we appreciate your expert opinion regarding the psychological well-being of survey respondents, which aligns with the first point in your peer review report concerning the anonymity component of cross-sectional surveys. Please review the revised "Study limitations" subsection of the discussion. Additionally, we acknowledge the possibility that some respondents, whether using or abstaining from psychedelics, may have preexisting psychopathologies or psychiatric comorbidities. However, screening or excluding such students would pose significant challenges and further limit survey response rates.

  • These could be suggestions for the implementation of the research in the future and not just criticisms of the current version of the work carried out.

Response: Dear Sir/Madam, we sincerely appreciate your time, efforts, and constructive critical analysis. We understand and value your aim to enhance the scholarly quality of the current study and future research in the field of psychedelic research.

Once more, we sincerely thank the respected reviewers for their dedicated efforts in reviewing our manuscript. We remain open to addressing additional concerns regarding our manuscript and are fully committed to meeting all specified requirements. Thank you for your continued consideration and support.

Best regards,

The corresponding author.

Reviewer 3 Report

Comments and Suggestions for Authors

The present manuscript reported the effects of culture on psychedelic awareness and use. The findings potentially have practical implications. However, there are several important issues.

1. Please clarify if the study was approved by the Institutional Review Board, or an equivalent institute.

2. While psychedelics overdose is an important public health issue, it is unclear why cultural differences in psychedelics use should be studied.

3. No hypotheses were clearly stated. The analyses were exploratory, and the findings were descriptive.

4. More details of sample recruitment are needed.  

Comments on the Quality of English Language

Minor English editing is required.

Author Response

Prof. Dr. José Jesús Gázquez Linares,

European Journal of Investigation in Health, Psychology and Education (EJIHPE),

Almería, Spain.

Poznan – Poland, 11th of February 2024

Subject: Revised Manuscript

Dear Prof. Dr. José Jesús Gázquez Linares, Editor-in-Chief of the EJIHPE Journal,

Dear Prof. Dr. María del Carmen Pérez-Fuentes, Editor-in-Chief of the EJIHPE Journal,

Dear Ms. Esther Liu, Managing Editor of the EJIHPE Journal,

             On behalf of all co-authors, we affirm our strict adherence to the valuable feedback the three esteemed reviewers provided. The revised version has garnered unanimous approval from all co-authors. Utilizing the track-changes option in Office Word, we edited the manuscript. We sincerely appreciate the review process's thoroughness and thank the journal's editors and reviewers for their dedicated efforts. Please find our responses to each comment below.

Reviewer-3

  • The present manuscript reported the effects of culture on psychedelic awareness and use. The findings potentially have practical implications. However, there are several important issues.

Response: Dear Sir/Madam, thank you for reviewing our manuscript. We thank you for acknowledging our study's practical applications and implications in Iraq and Poland, focusing on cultural differences. We have addressed the crucial issues you raised in your report.

  • Please clarify if the Institutional Review Board or an equivalent institute approved the study.

Response: Dear Sir/Madam, please check the "Research ethics" subsection of the methods and the "Institutional Review Board Statement" at the end of the manuscript. The study followed the ethical guidelines outlined in the Declaration of Helsinki by the World Medical Association and the Framingham Consensus of 1997. While no approval from the bioethics committee was necessary, it underwent examination by the Institutional Review Board (Ethics Committee) of Poznan University of Medical Sciences (Uniwersytet Medyczny im. Karola Marcinkowskiego w Poznaniu). It was exempted from approval requirements on the 12th of November, 2021, due to its observational nature (not experimental).

  • While psychedelic overdose is an important public health issue, it is unclear why cultural differences in psychedelic use should be studied.

Response: Dear Sir/Madam, thank you for raising this question. We have added the following answer to the "Study aims" subsection of the introduction. Psychedelics overdosing presents a significant public health concern in cultures worldwide, including Poland and Iraq, due to its associated health risks, such as psychosis and cardiovascular issues. Contributing factors include limited education on psychedelics' risks, varying legal statuses leading to misinformation and clandestine distribution, cultural attitudes affecting substance use perceptions and treatment-seeking behaviors, and disparities in access to healthcare and treatment services. Addressing this issue requires comprehensive approaches encompassing public education, harm reduction strategies, enhanced access to treatment, and culturally sensitive policies to effectively mitigate the impact of psychedelics overdosing across diverse cultural contexts.

  • No hypotheses were clearly stated. The analyses were exploratory, and the findings were descriptive.

Response: Dear Sir/Madam, thank you for your comment. Concerning the hypotheses, we revised the 1st paragraph of the study aims subsection of the introduction: "Our main goal is to assess awareness and usage of psychedelic substances among medical students from Poland and Iraq. This study aims to uncover differences in the prevalence and understanding of psychedelic substance use, shedding light on cultural, ethnic, religious, and educational factors influencing these behaviors. We hypothesize that Polish medical students have higher awareness and usage rates than their Iraqi counterparts.". The analyses rely on statistical inference and hypothesis testing using multivariable analyses through a binary logistic regression model. We reported odds ratios (OR) and 95% confidence intervals (95% CI). For further details, please refer to the "Statistical analysis and methods" section of the methods.

  • More details of sample recruitment are needed.

Response: Dear Sir/Madam, thank you for your insightful remark. We have revised the "Sampling and data collection" subsection of the methods. The survey was distributed to 3,768 students, comprising 1,306 Poles and 2,462 Iraqis. However, only 739 students responded and completed the survey, with 315 respondents from Poland and 424 from Iraq.

Once more, we sincerely thank the respected reviewers for their dedicated efforts in reviewing our manuscript. We remain open to addressing additional concerns regarding our manuscript and are fully committed to meeting all specified requirements. Thank you for your continued consideration and support.

Best regards,

The corresponding author.

Round 2

Reviewer 1 Report

Comments and Suggestions for Authors

I would like to thank authors for their value answers. Many of the explanation given are very suitable to be added along the manuscript, such as the explication for the "Iraqis tended to miss university classes deliberately" and “unrestricted access to the web” and table 3 explanation.

Author Response

Prof. Dr. José Jesús Gázquez Linares,

European Journal of Investigation in Health, Psychology and Education (EJIHPE),

Almería, Spain.

Poznan – Poland, 26th of February 2024

Subject: Revised Manuscript

Dear Prof. Dr. José Jesús Gázquez Linares, Editor-in-Chief of the EJIHPE Journal,

Dear Prof. Dr. María del Carmen Pérez-Fuentes, Editor-in-Chief of the EJIHPE Journal,

Dear Ms. Esther Liu, Managing Editor of the EJIHPE Journal,

             On behalf of all co-authors, we gratefully acknowledge the invaluable feedback provided by the respected reviewer, which we have incorporated into the revised version. Utilizing the track-changes feature in Office Word, we edited the manuscript. We sincerely appreciate the thoroughness of the review process and extend our thanks to the journal's editors and reviewers for their dedicated efforts.

Reviewer-1

  • I would like to thank the authors for their valuable answers. Many of the explanations given are very suitable to be added along the manuscript, such as the explication for the "Iraqis tended to miss university classes deliberately" and "unrestricted access to the web" and Table 3 explanation.

Response: Dear Sir/Madam, thank you for acknowledging our efforts in the initial peer review. We have revised the article following your comments, incorporating the highlighted explanations, including those in Table 3. We also added a new reference material (reference #3) concerning "The ambiguity of psychedelics use among Iraqi medical students". Additionally, we thoroughly proofread and double-checked the entire manuscript for accuracy.

Once more, we sincerely thank the respected reviewers for their dedicated efforts in reviewing our manuscript. We remain open to addressing additional concerns regarding our manuscript and are fully committed to meeting all specified requirements. Thank you for your continued consideration and support.

Best regards,

The corresponding author.
